# Association between sleep duration and chronic lung diseases among Chinese middle-aged and older adults: A cross-sectional study

Caixia Yang[1], Li Xu[2]*

1 Department of Statistics and Medical Record Management, Shandong Provincial Hospital Affiliated to Shandong First Medical University, Jinan, Shandong, China, 2 Department of Pulmonary and Critical Care Medicine, Shandong Provincial Hospital Affiliated to Shandong First Medical University, Jinan, Shandong, China

* 1111110236@bjmu.edu.cn

## Abstract

### Background

Chronic lung diseases (CLDs) continue to be a major global public health concern. Previous studies have produced inconsistent findings regarding the relationship between sleep duration and CLDs. This study aimed to examine the association between sleep duration and CLDs in Chinese middle-aged and elderly adults.

### Methods

This cross-sectional study used data from the 2011 survey of the China Health and Retirement Longitudinal Study (CHARLS). Self-reported sleep duration was collected using a structured questionnaire, and CLDs was defined by self-reported physician diagnosis. Multiple logistic regression models were employed to examine the association between sleep duration and CLDs. We used generalized additive model and smoothing fitted curves to examine whether nonlinear relationship existed.

### Results

A total of 13,759 participants were included. Among participants, 5,773 (42.0%) slept less than 7 hours, 4,711 (34.2%) slept between 7–8 hours, and 3,275 (23.8%) slept more than 8 hours. After adjusting for all covariates, logistic regression showed a positive association between short sleep duration (< 7 hours) and CLDs (OR 1.29, 95% CI 1.10–1.52) compared to the normal group (7–8 hours). In addition, smoothing fitted curves indicated the existence of a non-linear association between sleep duration and CLDs.

**Data availability statement:** The raw data for this study is available from: http://charls.pku.edu.cn/en. The dataset generated from this study can be obtained from the supporting information.

**Funding:** The author(s) received no specific funding for this work.

**Competing interests:** The authors have declared that no competing interests exist.

## Conclusions

Short sleep duration was associated with a higher likelihood of CLDs in Chinese middle-aged and elderly population. Further investigations are warranted to test this association.

## Introduction

Chronic lung diseases (CLDs) include a range of persistent conditions affecting the airways and lung structures [1]. Among these, chronic obstructive pulmonary disease (COPD) remains a major contributor to morbidity and mortality worldwide [1,2]. In China, the prevalence of spirometry-defined COPD is as high as 8.6% in the adult population [3], and the number of deaths due to COPD has reached 1,037,000 in 2019 [4]. While smoking is a well-established risk factor, other factors such as sleep duration, have gained increasing attention [5–9]. Therefore, it is imperative to deepen our understanding of the relationship between sleep duration and CLDs.

Sleep is a critical factor in strengthening the immune system and maintaining metabolic balance, memory consolidation, and brain detoxification [10–14]. CLDs and sleep may have a bidirectional relationship, with sleep quality and quantity influencing lung disease and vice versa [15]. Both inadequate and excessive sleep duration are related to a higher risk of respiratory diseases mortality [16]. Inflammatory responses may be involved in the potential mechanisms between abnormal sleep and lung disease. Studies showed that short or long sleep duration were related to higher levels of inflammatory cytokine, which may impair respiratory system [17–19].

However, the relationship between sleep duration and CLDs remains inconsistent across studies. Most of the evidence suggest that short sleep duration is associated with CLDs [5,8,9,20]. Studies in the U.S. Adults revealed that insufficient sleep was associated with COPD [8] and asthma [9]. Similarly, research from Jilin Province, China indicated an association between limited sleep and increased COPD odds [5,20]. In contrast, other studies reported conflicting findings. For examples, a study conducted in Suzhou City, China, indicated that long sleep duration might increase the risk for COPD, especially in adult smokers [21]. Another study of non-institutionalized Americans adults showed that both short and long sleep duration were associated with a higher odds of COPD [6]. Additionally, no association was observed between sleep duration and COPD in older Native Hawaiians or Other Pacific Islanders, in older Asian Americans [22], as well as among Chinese middle-aged and elderly adults [23]. The contradictory findings may stem from differences in study design, population characteristics, or statistical methods. And the conflicting findings indicate that further research is needed. Particularly in China, evidence for this topic based on the nationwide representative data is limited.

This study aimed to examine the association between sleep duration and CLDs among Chinese middle-aged and older adults based on the 2011 wave of the China Health and Retirement Longitudinal Study (CHARLS). We hypothesized that abnormal sleep durations was associated with CLDs.

## Methods

### Study population

CHARLS is a nationally representative longitudinal survey of Chinese adults aged 45 years or older. CHARLS uses a multistage stratified probability proportion to size sampling method to ensure a well-represented sample [24]. CHARLS conducted surveys across 150 counties and 450 communities (villages) in 28 provinces in 2011, 2013, 2015, 2018 and 2020, with follow-up waves every two to three years. The initial survey for CHARLS took place in 2011 and included data on demographics, living conditions, health status, physical measurements, other relevant information. Further details on the study are available on the CHARLS project website (http://charls.pku.edu.cn/).

CHARLS was approved by the Ethics Review Committee of Peking University. The ethical approval number is IRB00001052–11015. Written informed consent was obtained from all participants.

This cross-sectional study was an analytical study focusing on associations, not a descriptive study to obtain population estimates; therefore, sampling weights were not used. The study used data from the 2011 baseline survey, and included 17708 participants aged ≥ 45 years. The exclusion criteria were as follows: (1) age < 45 years, (2) missing CLDs, (3) missing sleep duration, (4) with asthma, (5) with emotional, nervous, or psychiatric problems, (6) with memory-related disorders, (7) certain missing covariates. We excluded participants with self-reported psychiatric and memory-related disorders due to potential recall bias. The selection process was fully illustrated in Fig 1. Ultimately, 13,759 participants were included in the study.

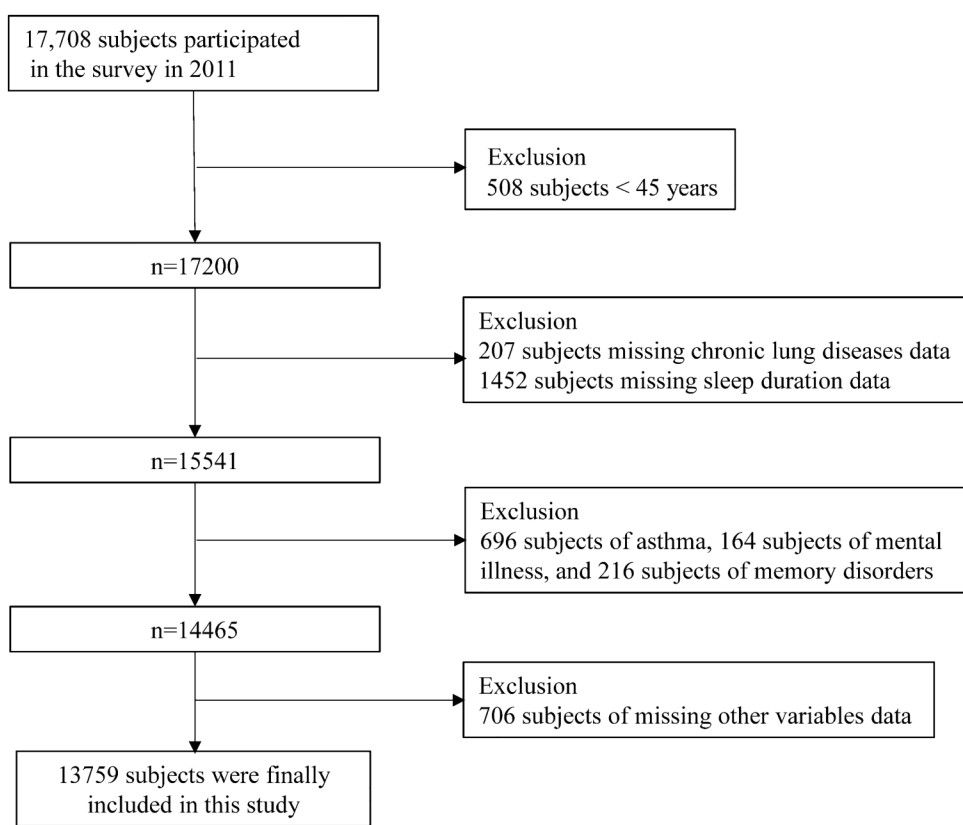

**Fig 1. Flow chart.**

## Assessment of sleep duration

Data on sleep duration were gathered through standardized questionnaires administered by trained interviewers. Nighttime sleep duration was assessed based on the question: "During the past month, how many hours of actual sleep did you get at night (average hours for one night)?". Nap time was assessed based on the question: "During the past month, how long did you take a nap after lunch (minutes)?". Sleep duration was calculated by adding nighttime sleep duration to nap time, with the nap time first divided by 60 to convert it into hours [5,25]. Based on this calculation, sleep duration was categorized as short sleep duration (< 7 hours), normal sleep duration (7–8 hours), or long sleep duration (> 8 hours) [5,8].

## Assessment of CLDs

Consistently with some references [23,26], CLDs were identified based on affirmative responses to the question: "Have you been diagnosed with chronic lung diseases, such as chronic bronchitis, emphysema (excluding tumors, or cancer) by a doctor?".

## Assessment of covariates

The selection of covariates was primarily based on several aspects: demographic characteristics, variables potentially associated with sleep duration and CLDs, and with reference to relevant literatures [5,8,20,25,26]. These covariates included age, gender, marital status, residence region, educational level, smoking status, drinking status, body mass index (BMI), cooking fuel, hypertension, diabetes, dyslipidemia, cardiovascular diseases, and depression.

The structured questionnaires collected sociodemographic and health-related information on age, gender, marital status, residence region, educational level, smoking status, drinking status, self-reported hypertension, diabetes, dyslipidemia, and cardiovascular diseases. Body weight, height were measured in light indoor clothes using a Seca™213 stadiometer, an Omron™ HN-286 scale, and BMI was calculated as weight in kilograms divided by height in meters squared (kg/m²) [24]. Cooking fuel obtained from household questionnaire was classified as solid fuel (a primary use of coal, crop residue or wood burning for cooking) or clean fuel (natural gas, marsh gas, liquefied petroleum gas, or electricity) [26]. Blood pressure was measured three times using the Omron™ HEM-7200 monitor after participants had rested for 30 minutes [24]. The average of the three readings was calculated and used. Hypertension was defined as self-reported physician diagnosis, or systolic blood pressure $\geq 140$ mmHg, or diastolic blood pressure $\geq 90$ mmHg [27,28]. Venous blood samples were collected by medical staff from the Chinese Center for Disease Control and Prevention and analyzed at the Youanmen Center Clinical Laboratory of Capital Medical University [29]. Diabetes was defined as self-reported diagnosis of diabetes, or a fasting plasma glucose level $\geq 126$ mg/dL, or glycosylated hemoglobin $\geq 6.5\%$ [30]. Dyslipidemia was defined as the self-reported diagnosis of dyslipidemia, a fasting plasma triglycerides $\geq 200$ mg/dL, total cholesterol $\geq 240$ mg/dL, high-density lipoprotein $< 40$ mg/dL, low-density lipoprotein $\geq 160$ mg/dL [31]. Cardiovascular diseases included heart diseases or stroke. Depressive symptoms were assessed by the 10-item Center for Epidemiologic Studies Depression scale (CESD-10), and depression was defined as having a CESD-10 score $\geq 10$ [32].

## Statistical analysis

Continuous variables were expressed as the mean $\pm$ standard deviation. Categorical variables were presented as the frequency (percentages). The differences between the groups (< 7 hours, 7–8 hours, > 8 hours) were compared using analysis of variance (for normal distribution) or Kruskal Wallis test (for skewed distribution) for continuous variables, and Chi-square test or Fisher exact test for categorical variables.

Multicollinearity among the variables was assessed using the variance inflation factor (VIF), and those with VIF value exceeding 10 were excluded as they were deemed collinear [33]. Univariate logistic regression model was used to examine the association between variables and CLDs. To investigate the association of sleep duration with CLDs, we employed

three logistic regression models. Model 1 was unadjusted. Model 2 was adjusted for age, gender, marital status, residence region, education level, BMI, smoke, drink. Model 3 was adjusted for all covariates. The selection of adjusted variables was primarily based on three key considerations: prior literature, clinical significance, and statistical results [5,8,20,25,26]. Generalized additive model with smoothing curve fitting were used to explore potenial nonlinear relationship. Threshold effect analysis was conducted using the two-piecewise regression model and the log likelihood ratio test [34,35].

Sensitivity analysis was performed as follows: (1) subgroup analyses were conducted using logistic regression model adjusting for different covariates except effect modifier to examine whether the association between sleep duration and CLDs was moderated by the following characteristics: age, gender, educational level, residence region, BMI, smoking status, cooking fuel. Interaction tests were evaluated using likelihood ratio test. (2) repeating main regression analyses after the multiple imputation method (based on 5 replications in the R MICE procedure) for addressing missing variables.

The missing data for BMI was16.03%, and we used dummy variable to indicate missing values. The missing samples were recorded as a group and marked as "unclear".

Data sorting was carried out using STATA software (version 15.0). All analyses were conducted with Empower software (www.empowerstats.com, X&Y solutions, Inc. Boston, Massachusetts) and R version 4.2.0 (http://www.R-project.org, The R Foundation). A two-tailed $P<0.05$ was considered statistically significant.

## Results

### Baseline characteristics

The study included 13,759 participants (Fig 1). As demonstrated in Table 1, 5,773 (42.0%) individuals slept less than 7 hours, 4,711 (34.2%) slept between 7–8 hours, and 3,275 (25.8%) slept more than 8 hours. Age, gender, marital status, residence region, education level, smoking status, drinking status, BMI, cooking fuel, hypertension, cardiovascular diseases, and depression all exhibited statistically significant differences across the groups ($P<0.05$). While, diabetes and dyslipidemia were not statistically significant across the groups ($P>0.05$). Participants in the short sleep duration group were more likely to have CLDs (8.5%), and CLDs was statistically significant across the groups ($P<0.001$).

### Association between sleep duration and CLDs

The VIF values were all below 10, and covariates were entered the fully-adjusted model. Univariate analysis showed that sleep duration was statistically associated with CLDs (S1 Table). Table 2 illustrated the association between sleep duration and CLDs. When treated as continuous variable, each additional hour of sleep was associated with a 8% decrease in the prevalence of CLDs (OR 0.92, 95% CI 0.89–0.95) in the model 1; each additional hour of sleep was associated with a 7% decrease in the prevalence of CLDs (OR 0.93, 95% CI 0.90–0.96) in the model 2; each additional hour of sleep was associated with a 4% decrease in the prevalence of CLDs (OR 0.96, 95% CI 0.93–0.99) in the model 3 adjusted for all covariates. As categorical variable, with the reference of normal sleep duration, short sleep duration was positively associated with CLDs (model 1: OR 1.52, 95% CI 1.31–1.78; model 2: OR 1.44, 95% CI 1.23–1.69; model 3: OR 1.29, 95% CI 1.10–1.52). However, long sleep duration was not significantly associated with CLDs (OR 1.18, 95% CI 0.98–1.42) after adjustment for all covariates.

A non-linear relationship between sleep duration and CLDs was depicted in Fig 2. With an increase in sleep duration, the probability of CLDs decreased, plateauing at approximately 7–8 hours of sleep, where the probability of CLDs no longer changed. The log likelihood ratio test indicated that the inflection points of 7 and 8 hours existed (S2 Table).

### Sensitivity analyses

We executed subgroup analyses and interaction tests, stratified by age, gender, education level, residence region, BMI, smoking status, and cooking fuel. The results were summarized as follows (Table 3): short sleep duration was

**Table 1. Basic characteristics of participants.**

| Variables | Sleep duration (h) | | | P |
|---|---|---|---|---|
| | 7-8 (n = 4711) | < 7 (n = 5773) | > 8 (n = 3275) | |
| Age (years) | 57.8±9.1 | 60.1±9.5 | 59.3±9.9 | < 0.001 |
| Age groups | | | | < 0.001 |
| <60 | 2917 (61.9%) | 3025 (52.4%) | 1837 (56.1%) | |
| ≥60 | 1794 (38.1%) | 2748 (47.6%) | 1438 (43.9%) | |
| Gender | | | | < 0.001 |
| Female | 2354 (50.0%) | 3358 (58.2%) | 1551 (47.4%) | |
| Male | 2357 (50.0%) | 2415 (41.8%) | 1724 (52.6%) | |
| Marital status | | | | < 0.001 |
| Married/living with partner | 4234 (89.9%) | 4932 (85.4%) | 2908 (88.8%) | |
| Other | 477 (10.1%) | 841 (14.6%) | 367 (11.2%) | |
| Residence region | | | | < 0.001 |
| Urban | 2073 (44.0%) | 2274 (39.4%) | 1262 (38.5%) | |
| Rural | 2638 (56.0%) | 3499 (60.6%) | 2013 (61.5%) | |
| Education level | | | | < 0.001 |
| <Primary school | 1827 (38.8%) | 2936 (50.9%) | 1426 (43.5%) | |
| Primary school | 1044 (22.2%) | 1155 (20.0%) | 701 (21.4%) | |
| Middle school | 1080 (22.9%) | 1086 (18.8%) | 727 (22.2%) | |
| ≥High school | 760 (16.1%) | 596 (10.3%) | 421 (12.9%) | |
| Smoking status | | | | < 0.001 |
| Non-smoker | 2813 (59.7%) | 3704 (64.2%) | 1903 (58.1%) | |
| Smoker | 1509 (32.0%) | 1624 (28.1%) | 1063 (32.5%) | |
| Former smoker | 389 (8.3%) | 445 (7.7%) | 309 (9.4%) | |
| Drinking Status | | | | < 0.001 |
| More than once a month | 1248 (26.5%) | 1333 (23.1%) | 874 (26.7%) | |
| Less than once a month | 365 (7.7%) | 430 (7.4%) | 269 (8.2%) | |
| Non-drinker | 3098 (65.8%) | 4010 (69.5%) | 2132 (65.1%) | |
| Body mass index, kg/m$^2$ | | | | < 0.001 |
| <24 | 2307 (49.0%) | 3007 (52.1%) | 1586 (48.4% | |
| ≥24 | 1690 (35.9%) | 1804 (31.2%) | 1160 (35.4%) | |
| unclear | 714 (15.2%) | 962 (16.7%) | 529 (16.2%) | |
| Cooking fuel | | | | < 0.001 |
| Clean fuel | 2395 (50.8%) | 2560 (44.3%) | 1466 (44.8%) | |
| Solid fuel | 2316 (49.2%) | 3213 (55.7%) | 1809 (55.2%) | |
| Hypertension | | | | 0.002 |
| Yes | 1725 (36.6%) | 2275 (39.4%) | 1310 (40.0%) | |
| No | 2986 (63.4%) | 3498 (60.6%) | 1965 (60.0%) | |
| Diabetes | | | | 0.196 |
| Yes | 560 (11.9%) | 702 (12.2%) | 432 (13.2%) | |
| No | 4151 (88.1%) | 5071 (87.8%) | 2843 (86.8%) | |
| Dyslipidemia | | | | 0.096 |
| Yes | 1429 (30.3%) | 1737 (30.1%) | 1054 (32.2%) | |
| No | 3282 (69.7%) | 4036 (69.9%) | 2221 (67.8%) | |
| Cardiovascular diseases | | | | < 0.001 |
| Yes | 540 (11.5%) | 897 (15.5%) | 392 (12.0%) | |
| No | 4171 (88.5%) | 4876 (84.5%) | 2883 (88.0%) | |

*(Continued)*

**Table 1.** (Continued)

| Variables | Sleep duration (h) | | | P |
|---|---|---|---|---|
| | 7-8 (n = 4711) | < 7 (n = 5773) | > 8 (n = 3275) | |
| Depression | | | | < 0.001 |
| Yes | 1242 (26.4%) | 2778 (48.1%) | 851 (26.0%) | |
| No | 3469 (73.6%) | 2995 (51.9%) | 2424 (74.0%) | |
| Chronic lung diseases | | | | < 0.001 |
| Yes | 271 (5.8%) | 491 (8.5%) | 234 (7.1%) | |
| No | 4440 (94.2%) | 5282 (91.5%) | 3041 (92.9%) | |

**Table 2. Logistic regression analysis between sleep duration and chronic lung diseases.**

| Variables | Model 1 | P | Model 2 | P | Model 3 | P |
|---|---|---|---|---|---|---|
| | OR (95% CI) | | OR (95% CI) | | OR (95% CI) | |
| Sleep duration (h) | 0.92 (0.89, 0.95) | < 0.001 | 0.93 (0.90, 0.96) | < 0.001 | 0.96 (0.93, 0.99) | 0.004 |
| Sleep duration (h) | | | | | | |
| 7-8 | Reference | | Reference | | Reference | |
| < 7 | 1.52 (1.31, 1.78) | < 0.001 | 1.44 (1.23, 1.69) | < 0.001 | 1.29 (1.10, 1.52) | 0.002 |
| > 8 | 1.26 (1.05, 1.51) | 0.012 | 1.18 (0.99, 1.42) | 0.072 | 1.18 (0.98, 1.42) | 0.072 |

Note: Model 1 adjusted for nothing. Model 2 adjusted for age, gender, marital status, education level, residence region, BMI, smoke, drink. Model 3 adjusted for model 2 plus cooking fuel, hypertension, diabetes, dyslipidemia, cardiovascular disease and depression.

associated with CLDs in age group < 60 years (OR 1.26, 95% CI 1.00–1.60), in age group ≥ 60 years (OR 1.33, 95% CI 1.07–1.65), in the male group (OR 1.30, 95% CI 1.05–1.61), in urban dwellers (OR 1.44, 95% CI 1.11–1.86), in rural dwellers (OR 1.27, 95% CI 1.03–1.55), in BMI < 24 group (OR 1.31, 95% CI 1.06–1.61), in BMI ≥ 24 group (OR 1.44, 95% CI 1.08–1.93), in the smoking group (OR 1.37, 95% CI 1.05–1.79), in solid fuels group (OR 1.33, 95% CI 1.07–1.64). The relationship between sleep duration and CLDs was consistent across all subgroups (all P for interaction > 0.05).

In addition, the results of multivariable regression analysis after multiple imputation for missing data showed short sleep duration was associated with CLDs (OR 1.29, 95% CI 1.11–1.51) (S3 Table).

## Discussion

In this cross-sectional study, short sleep duration (< 7 hours) was associated with a higher likelihood of CLDs (OR 1.29, 95% CI 1.10–1.52) compared to the reference group (7–8 hours) after adjusting for potential confounding factors in Chinese middle-aged and elderly population. A non-linear relationship between sleep duration and CLDs was found. To our knowledge, this is the first study to comprehensively examine the association between sleep duration and CLDs in Chinese adults aged 45 and over.

Sleep is a vital physiological process. Adequate sleep not only enhances daily cognitive and physical performance but also boosts immune function, contributing to overall health [10,13]. Multiple studies showed the links between insufficient sleep and chronic diseases such as obesity, type 2 diabetes, hypertension, and cardiovascular diseases [36–38]. Sleep deprivation induces immune activation characterized by elevated levels of pro-inflammatory cytokines (IL-1β, TNF-α, and IL-6) and serum immunoglobulins (IgM, IgG, and IgA) [39–41], which may worsen respiratory system health [18,42]. Our findings further provided epidemiological evidence that short sleep duration was associated with a higher likelihood of CLDs.

 

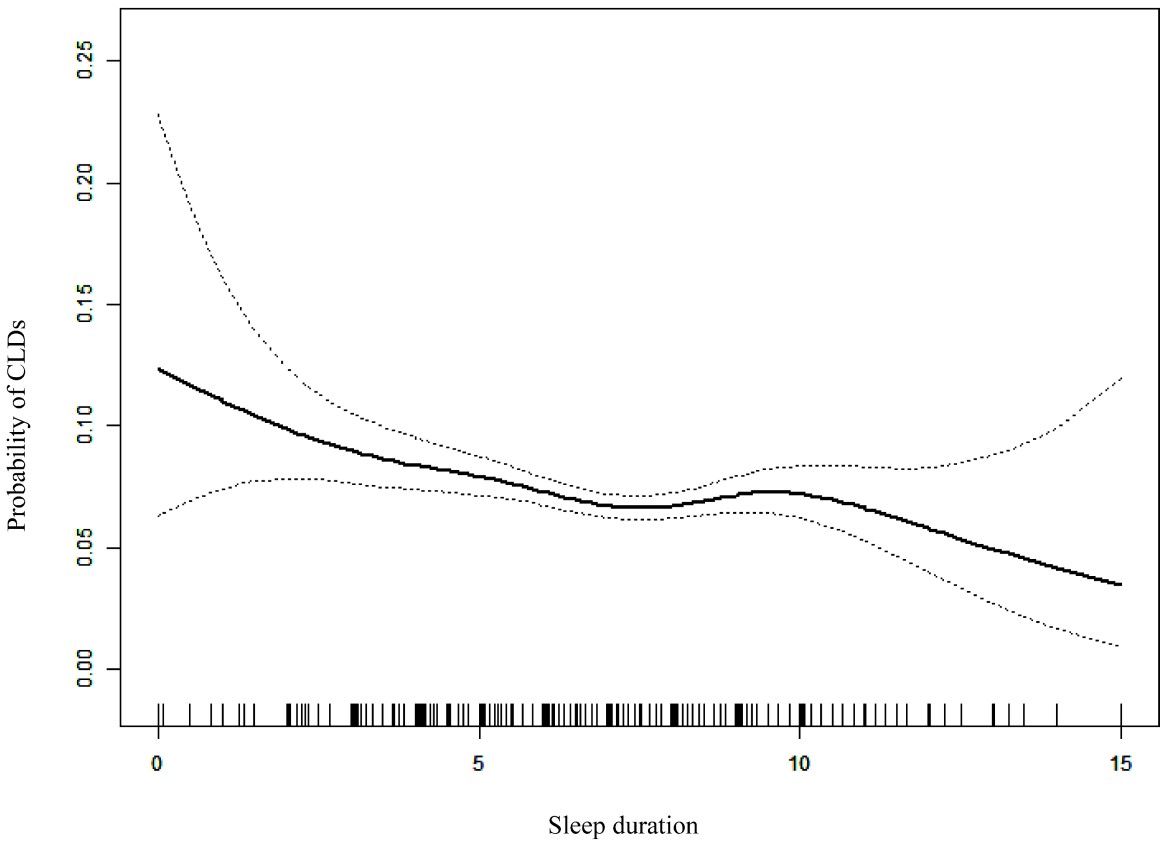

**Fig 2. Dose-response relationship between sleep duration and CLDs.** Solid black line represents the smooth curve fit between variables. Dotted lines bands represent the 95% of confidence interval adjusted for all covariates. CLDs: chronic lung diseases.

Research on the association of sleep duration with respiratory health has garnered increasing attention. A cross-sectional research of 14,742 adults aged ≥ 40 years in the United States, utilizing data from the National Health and Nutrition Examination Survey from 2005 to 2012, identified short sleep duration of less than 7 hours exhibited a significant association with COPD [8]. Similar findings were reported that short sleep duration were positively associated with COPD in two cross-sectional investigations involving both 4,115 adults aged 60–79 years and17,320 adults aged 18–59 years in Jilin Province, China [5,20]. Consistently with the above studies, our findings showed that short sleep duration was positively associated with CLDs in the Chinese middle-aged and elderly adults.

However, several studies reported divergent findings regarding the association between sleep duration and CLDs. For instance, a cross-sectional study involving 1,191,768 American adults aged ≥18 years, utilizing data from the Behavioral Risk Factor Surveillance System of 2013, 2014 and 2016, found a marked positive association between both very short and very long sleep duration and the odds of COPD [6]. A prospective study of 53,269 participants aged 30–79 years in Suzhou City, China, suggested that prolonged sleep duration was associated with an increased risk of COPD [21]. A longitudinal cohort study used data from the CHARLS including 7,025 Chinese participants aged ≥ 45 years, found no significant association between sleep duration and CLDs [23]. Additionally, the findings among 1127 Native Hawaiians or Other Pacific Islanders and 4655 Asian Americans adults ≥ 50 years, were also found no link between sleep duration and respiratory disease using the data of the 2016 Behavioral Risk Factor Surveillance System [22].

**Table 3. Subgroup analysis between sleep duration and chronic lung diseases.**

| Subgroup | Sleep duration (h) | Adjusted OR (95% CI) | P for interaction |
|---|---|---|---|
| Age (years) | | | 0.924 |
| <60 | 7-8 | Reference | |
| | < 7 | 1.26 (1.00, 1.60) | |
| | > 8 | 1.14 (0.87, 1.50) | |
| ≥60 | 7-8 | Reference | |
| | < 7 | 1.33 (1.07, 1.65) | |
| | > 8 | 1.22 (0.96, 1.57) | |
| Gender | | | 0.984 |
| Female | 7-8 | Reference | |
| | < 7 | 1.26 (0.99, 1.60) | |
| | > 8 | 1.19 (0.89, 1.59) | |
| Male | 7-8 | Reference | |
| | < 7 | 1.30 (1.05, 1.61) | |
| | > 8 | 1.18 (0.93, 1.50) | |
| Education level | | | 0.262 |
| <Primary school | 7-8 | Reference | |
| | < 7 | 1.21 (0.96, 1.52) | |
| | > 8 | 1.30 (0.99, 1.69) | |
| Primary school | 7-8 | Reference | |
| | < 7 | 1.63 (1.17, 2.27) | |
| | > 8 | 1.14 (0.77, 1.70) | |
| Middle school | 7-8 | Reference | |
| | < 7 | 1.21 (0.85, 1.74) | |
| | > 8 | 0.97 (0.63, 1.48) | |
| ≥ High school | 7-8 | Reference | |
| | < 7 | 1.22 (0.73, 2.05) | |
| | > 8 | 1.05 (0.60, 1.87) | |
| Residence region | | | 0.312 |
| Urban | 7-8 | Reference | |
| | < 7 | 1.44 (1.11, 1.86) | |
| | > 8 | 1.09 (0.80, 1.49) | |
| Rural | 7-8 | Reference | |
| | < 7 | 1.27 (1.03, 1.55) | |
| | > 8 | 1.27 (1.01, 1.60) | |
| Body mass index, kg/m$^2$ | | | 0.322 |
| <24 | 7-8 | Reference | |
| | < 7 | 1.31 (1.06, 1.61) | |
| | > 8 | 1.25 (0.98, 1.59) | |
| ≥24 | 7-8 | Reference | |
| | < 7 | 1.44 (1.08, 1.93) | |
| | > 8 | 0.98 (0.69, 1.39) | |
| Unclear | 7-8 | Reference | |
| | < 7 | 1.71 (1.12, 2.63) | |
| | > 8 | 1.36 (0.83, 2.23) | |
| Smoking status | | | 0.923 |
| Non-smoker | 7-8 | Reference | |

*(Continued)*

**Table 3.** (Continued)

| Subgroup | Sleep duration (h) | Adjusted OR (95% CI) | P for interaction |
|---|---|---|---|
| | < 7 | 1.24 (0.99, 1.55) | |
| | > 8 | 1.22 (0.94, 1.58) | |
| Smoker | 7-8 | Reference | |
| | < 7 | 1.37 (1.05, 1.79) | |
| | > 8 | 1.12 (0.83, 1.52) | |
| Ex-smoker | 7-8 | Reference | |
| | < 7 | 1.34 (0.86, 2.10) | |
| | > 8 | 1.26 (0.77, 2.06) | |
| Cooking fuel | | | 0.801 |
| Clean fuels | 7-8 | Reference | |
| | < 7 | 1.26 (0.98, 1.61) | |
| | > 8 | 1.08 (0.81, 1.45) | |
| Solid fuels | 7-8 | Reference | |
| | < 7 | 1.33 (1.07, 1.64) | |
| | > 8 | 1.26 (0.99, 1.60) | |

Note: Adjusted for all covariates except effect modifier.

These inconsistent findings across studies can be explained by several critical factors. First, the representativeness of the studied population must be considered. Demographic differences, including variations in ethnicity and age distribution, could potentially influence the study results. Second, different studies used varying methods to define sleep duration and categorize it as insufficient, normal, or excessive, which may affect the results. Additionally, whether the duration of sleep includes napping time may also influence the results. Finally, the degree of control for potential confounders also may led to differences. It is important to recognize that all studies did not uniformly adjust for factors such as socioeconomic status, underlying health conditions, or psychological variables. Our research specifically adjusted for several pertinent factors, such as cooking fuel types and depression, which may influence respiratory health outcomes and sleep duration [26,33].

Our study provided a reference for incorporating sleep management into the management of CLDs, highlighting the need for public health professionals to pay greater attention to sleep health in middle-aged and elderly population. We suggest the following measures to improve sleep health: raising awareness of sleep health through community-based sleep hygiene education programs, conducting sleep problem screening in primary healthcare institutions, and establishing multidisciplinary sleep clinics. Potential barriers may include limited healthcare resources and socio-cultural attitudes toward sleep. We recommend establishing a collaborative mechanism among healthcare providers, policymakers, and community organizations, and modifying socio-economic and cultural norms to promote sleep health.

This study had several strengths. Our research originated from high-quality national cohort study, and included and controlled for a considerable number of possible confounding factors such as cooking fuel and depression. Additionally, the consistency of more subgroup analyses and interaction tests indicated the robustness of our results. Our study explored the association between total sleep duration within a 24-hour period and CLDs, and excluded the potential influence of nap time on nighttime sleep. However, our study has some limitations. First, the cross-sectional study can not infer causal relationships. Because, it simultaneously measure exposure and outcome variables, and are unable to determine the temporal sequence between them, which raises the possibility of reverse causality. Additionally, residual confounding by unmeasured variables inevitably exist in cross-sectional studies that may bias the observed association. Second, sleep duration measures in our study were self-reported by participants and were subject to recall bias. Although

subjectively reported sleep duration modestly correlated with objectively sleep duration [43], more precise techniques such as polysomnography are still needed for future study. Third, CLDs are assessed through self-reported physician diagnosis without verification from medical records or diagnostic tests. This approach might introduce the potential for misclassification bias. Future studies should incorporate objective measures to improve the accuracy and reliability of disease diagnosis. Lastly, our study conducted in participants aged ≥ 45 years in China, this findings do not generalize to those younger than 45 years of age or other countries.

## Conclusion

Our study revealed that short sleep duration was associated with a higher likelihood of CLDs in Chinese middle-aged and elderly population. Future research should focus on prospective cohort studies to establish the causal relationship between sleep duration and CLDs, the interaction between sleep quality and sleep duration, and the association of dynamic changes in sleep patterns with CLDs.

## Supporting information

**S1 Table. Results of univariate analysis.**
(XLSX)

**S2 Table. Threshold effect analysis for the relationship between sleep duration and CLDs.**
(XLSX)

**S3 Table. Logistic regression analysis between sleep duration and chronic lung diseases after multiple imputation.**
(XLSX)

## Author contributions

**Conceptualization:** Caixia Yang, Li Xu.

**Data curation:** Caixia Yang.

**Writing – original draft:** Caixia Yang.

**Writing – review & editing:** Caixia Yang, Li Xu.

## Acknowledgments

We extend our deepest appreciation to Peking University for giving access to CHARLS data, and to all individuals partaking in both data accumulation and administration.

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
