## [Decision Letter · Decision Letter 0]

26 Dec 2024

PONE-D-24-40498

Association between sleep duration and chronic lung diseases among Chinese middle-aged and older adults: a cross-sectional study

Dear Dr. yang,

Please submit your revised manuscript by Feb 09 2025 11:59PM. If you will need more time than this to complete your revisions, please reply to this message or contact the journal office at plosone@plos.org . A rebuttal letter that responds to each point raised by the academic editor and reviewer(s). You should upload this letter as a separate file labeled 'Response to Reviewers'.A marked-up copy of your manuscript that highlights changes made to the original version. You should upload this as a separate file labeled 'Revised Manuscript with Track Changes'.An unmarked version of your revised paper without tracked changes. You should upload this as a separate file labeled 'Manuscript'.

We look forward to receiving your revised manuscript.

Kind regards,

Gebrekidan Ewnetu Tarekegn, MPH, MSc Academic Editor

PLOS ONE

2. In the online submission form you indicate that your data is not available for proprietary reasons and have provided a contact point for accessing this data. Please note that your current contact point is a co-author on this manuscript. According to our Data Policy, the contact point must not be an author on the manuscript and must be an institutional contact, ideally not an individual. Please revise your data statement to a non-author institutional point of contact, such as a data access or ethics committee, and send this to us via return email. Please also include contact information for the third party organization, and please include the full citation of where the data can be found.

6. Please include your tables as part of your main manuscript and remove the individual files. Please note that supplementary tables (should remain/ be uploaded) as separate "supporting information" files.

7. We noticed you have some minor occurrence of overlapping text with the  previous publication(s), which needs to be addressed:

**Comments to the Author**

1. Is the manuscript technically sound, and do the data support the conclusions?

Reviewer #1: Partly

Reviewer #2: Yes

2. Has the statistical analysis been performed appropriately and rigorously?

Reviewer #1: Yes

Reviewer #2: No

3. Have the authors made all data underlying the findings in their manuscript fully available?

Reviewer #1: Yes

Reviewer #2: Yes

4. Is the manuscript presented in an intelligible fashion and written in standard English?

Reviewer #1: Yes

Reviewer #2: Yes

Reviewer #1: I have added comments in response to those questions. For some un clear technical issues and models that used in this study, I made a suggestions and comments for the author. The data is available online and permission was also given

Reviewer #2: Detailed Critique:

1. Introduction:

o Contextual Background: The introduction broadly discusses the prevalence of chronic diseases among older adults in China and the importance of non-smoking risk factors, such as sleep duration. However, it lacks a clear, concise statement of the specific gap in research that this study aims to fill. Clarifying this could help strengthen the rationale for the study.

o Literature Review: The review of existing studies on sleep duration and chronic lung diseases appears somewhat cursory and could benefit from a more detailed discussion. Including a more thorough analysis of previous findings, especially contrasting studies with conflicting results, would provide a stronger foundation for the study's objectives.

o Link between Sleep and Lung Health: While the introduction mentions that sleep influences lung health, it could delve deeper into the physiological or biochemical pathways that might mediate this relationship. Providing a brief overview of potential mechanisms could help readers understand why sleep duration might impact chronic lung disease.

o Epidemiological Data: The introduction cites several statistics regarding chronic diseases and mortality but does not clearly connect these data points specifically to lung health or elaborate on how these figures relate to the study’s focus on sleep duration. A more direct connection between these statistics and the study’s aims would enhance the introduction’s relevance.

o Clarity and Focus: The introduction might benefit from tighter editing to improve clarity and focus. Some sentences are overly complex and could be simplified to improve readability and ensure that the study’s aims are immediately clear to the reader.

o Hypothesis or Research Questions: The introduction does not clearly state the research questions or hypotheses. Explicitly stating what the study intends to examine would help to frame the research context and provide a clearer guide for the reader through the rest of the document.

2. Methods:

Study Design and Population Description:

o The manuscript states that CHARLS is a prospective longitudinal study, yet the study design used for this particular analysis is cross-sectional. This should be clarified to avoid confusion regarding the nature of the data analysis.

o The inclusion criteria and study population are described, but more detailed justification on the age cut-off and criteria selection could provide better understanding and rationale for their choices.

Measurement of Sleep Duration:

o The method of measuring sleep duration combines night sleep and naps without discussing potential biases or the validity of this approach. The impact of napping on overall sleep duration and its potential differential impact on health outcomes needs further discussion or acknowledgment as a limitation.

o Only providing average sleep over the past month may not accurately reflect habitual patterns or variability in sleep, which can be crucial in chronic disease frameworks.

Assessment of Chronic Lung Diseases (CLDs):

o CLDs are assessed solely through self-reporting without verification from medical records or diagnostic tests, which could lead to misclassification bias. This limitation is significant and should be addressed more comprehensively in the discussion of the methodological limitations.

Covariate Assessment:

o The selection of covariates is appropriate; however, the manuscript could benefit from a more detailed explanation of why these particular covariates were chosen over others. Additionally, the methods for measuring these variables (e.g., self-report vs. clinical measurement for hypertension and diabetes) need clarification.

o The handling of missing data by deletion or dummy variable method may introduce bias, especially if the data are not missing completely at random. More sophisticated methods such as multiple imputation could be considered and discussed.

Statistical Analysis:

o The use of logistic regression models is appropriate; however, the decision to use a three-piecewise linear regression model to explore non-linear relationships is briefly mentioned without sufficient justification for this choice. A detailed rationale for selecting this model would strengthen the methodology.

o The manuscript would benefit from additional information on the model fit, diagnostic checks, and whether the assumptions of the used statistical models were met.

Subgroup Analysis and Interaction Tests:

o While subgroup analyses add depth, there is a risk of Type I errors due to multiple comparisons, which does not appear to be addressed.

o Interaction tests are mentioned, but the criteria for choosing these particular interactions are not discussed. Additionally, the manuscript should clarify how these interactions were tested statistically and interpret their clinical implications.

Software and Statistical Significance:

o Mentioning the use of STATA and R is helpful, but specific packages or commands used could be included for reproducibility.

o The manuscript declares a p-value of lss than 0.05 as statistically significant without discussion on the correction for multiple testing, which could be important given the number of analyses conducted.

3. Results:

o Cross-sectional Design Limitations: The use of a cross-sectional design limits the ability to infer causality between sleep duration and chronic lung diseases (CLDs). The authors do not sufficiently address this inherent limitation in their methodology section or discuss the potential for reverse causation—whereby individuals with CLDs might experience disturbed sleep due to their condition.

o Measurement of Sleep Duration: Sleep duration is self-reported, which introduces potential recall bias and misclassification of the exposure variable. The authors do not discuss how this might impact the reliability of their findings or any measures taken to mitigate this bias.

o Assessment of Chronic Lung Diseases: CLDs are identified solely based on participant self-report without confirmation by medical records or clinical evaluation. This could lead to misdiagnosis or under-reporting of the condition, significantly affecting the study's outcome measurements.

o Confounding Variables: While the authors adjust for a variety of potential confounders in their regression models, there is inadequate discussion regarding the selection and exclusion of these variables. The possibility of residual confounding by unmeasured or inadequately measured variables (e.g., environmental pollutants, occupational exposures) is not addressed.

o Subgroup Analyses: The manuscript reports multiple subgroup analyses without adjusting for multiple comparisons, which increases the risk of Type I errors. The criteria for choosing these particular subgroups for analysis are not well-explained, which might suggest data dredging.

o Statistical Methods and Model Fit: The use of a three-piecewise linear regression model to determine inflection points is not sufficiently justified with theoretical or empirical rationale. Furthermore, the authors do not provide diagnostics to evaluate the fit of this model or the logistic regression models used.

o Generalizability: The findings are based on a sample from the China Health and Retirement Longitudinal Study (CHARLS), and the authors claim national representativeness. However, they do not thoroughly discuss the potential limitations in generalizing these findings to other populations or settings outside China.

o Data Availability: The data are said to be available upon request, which could restrict access for some researchers. The authors do not discuss why the data cannot be deposited in a public repository, which might limit transparency and reproducibility of the research.

o Ethical Considerations: The study involves human participants, and while it mentions ethical approval, there is little discussion on the consent process, particularly how informed consent was obtained from participants or if any special considerations were made for vulnerable populations like the elderly.

o Potential Bias in Authorship: The study lists only one author, which could raise concerns about the breadth of expertise and potential biases in conducting the research and interpreting the findings. Collaborative research often benefits from diverse perspectives, particularly in multidisciplinary fields like epidemiology and public health.

4. Discussion:

o Causal Inferences: The authors discuss the results as if they imply causality between sleep duration and chronic lung diseases (CLDs). Given the cross-sectional nature of the study, such inferences are inappropriate as they cannot establish causality due to the simultaneous measurement of exposure and outcome. This limitation should be acknowledged more robustly, with a clearer statement that the findings can only suggest association, not causation.

o Comparison with Other Studies: While the discussion mentions various studies that either support or contradict the findings, there is a lack of critical analysis regarding why these differences might exist. The authors could enhance the discussion by exploring potential reasons for discrepancies, such as differences in study design, population characteristics, or methods of measuring sleep duration and lung health.

o Generalizability: The authors claim that their findings are representative of the broader Chinese population aged 45 and over. However, they do not adequately address the potential for selection bias or the impact of regional variations within China that might affect the generalizability of the results. A more detailed discussion on the representativeness of the sample and its implications for applying these findings to different subgroups or geographic areas would be beneficial.

o Mechanistic Insights: The discussion provides a general overview of the physiological mechanisms by which sleep might influence lung health, such as through inflammation and immune function. However, these explanations are somewhat superficial and generic. The section would benefit from a more detailed, specific discussion of the mechanisms potentially linking sleep duration with the specific types of chronic lung disease observed, possibly incorporating recent research or theoretical models.

o Methodological Limitations: The discussion touches on the limitations of the study but does not critically engage with how these might have influenced the results. For instance, the reliance on self-reported sleep and disease measures could introduce significant bias, which is not thoroughly analyzed. Discussing the potential impact of these biases and how they might be mitigated in future research would strengthen the section.

o Implications for Public Health: While the discussion suggests potential public health interventions, it does so without a clear strategy for implementation or consideration of the challenges that might be faced in promoting sleep health among middle-aged and older adults. A more nuanced discussion on the practical steps, potential barriers, and necessary collaborations for effective intervention would provide valuable insights for readers.

o Future Research Directions: The section could be improved by offering more concrete suggestions for future research, such as longitudinal studies or clinical trials that could validate the findings and provide stronger evidence for causality. Additionally, exploring other relevant factors such as sleep quality and its interaction with sleep duration could be suggested.

o

5. Limitations:

o Self-Reported Measures: The study heavily relies on self-reported data for both sleep duration and the diagnosis of chronic lung diseases. Self-reported data can be subject to recall bias and misclassification, which might affect the study's accuracy and reliability. Discussing the potential impact of these biases and how they might affect the findings would provide a more comprehensive view of the study's limitations.

o Cross-Sectional Design: While you briefly mention the inability to establish causality due to the cross-sectional design, this limitation deserves more emphasis. It's crucial to discuss how this design restricts the ability to discern whether reduced sleep actually leads to increased incidence of chronic lung diseases or if perhaps individuals with such diseases might experience sleep disturbances due to their condition.

o Lack of Polysomnography Data: The study uses a simplified measure of sleep duration without validation through more objective measures such as polysomnography, which can provide detailed insights into sleep quality and architecture. Mentioning the lack of such data as a limitation would be beneficial, as it could affect the interpretation of the relationship between sleep quality and lung health.

o Generalizability: The findings are based on a specific demographic (Chinese middle-aged and older adults), which may not be generalizable to other populations or age groups. Expanding on the limitations in terms of ethnic and demographic diversity could highlight the need for caution when extrapolating these results to other populations.

o Confounding Variables: While the study adjusts for numerous potential confounders, there might be unmeasured or residual confounding factors such as environmental or occupational exposures that can influence both sleep and lung health. More discussion on the possibility of such unmeasured confounders would strengthen the limitations section.

o Temporal Fluctuations in Sleep Patterns: The study does not account for changes in sleep patterns over time, which can be particularly relevant in an aging population. The inability to track changes in sleep duration over time and its potential impact on the development of chronic conditions is a notable limitation.

o Details on Data Collection Methods: More detailed criticism could be levied at the methods used for data collection, particularly if the sleep data collection did not differentiate between weekdays and weekends or consider variability in sleep patterns, which could influence the overall sleep duration reported.

o

6. Ethical Considerations:

o The ethical approval and consent procedures are adequately described. However, a statement about how the data were anonymized or any other data security measures taken during the analysis would enhance the ethical reporting.

7. Conclusion:

o The conclusion succinctly summarizes the findings, but it could better highlight the practical implications of the research, suggesting specific public health interventions or policy changes that could be informed by the study results.

**Do you want your identity to be public for this peer review?** For information about this choice, including consent withdrawal, please see our Privacy Policy

Reviewer #1: No

Reviewer #2: **Yes:** Endalkachew Belayneh Melese

---

## [Decision Letter · Decision Letter 1]

12 May 2025

Dear Dr. Xu,

Thank you for submitting your manuscript to PLOS ONE. After careful consideration, we feel that it has merit but does not fully meet PLOS ONE’s publication criteria as it currently stands. Therefore, we invite you to submit a revised version of the manuscript that addresses the points raised during the review process.

We look forward to receiving your revised manuscript.

Kind regards,

Bisher Sawaf

Academic Editor

PLOS ONE

Reviewers' comments:

Reviewer's Responses to Questions

**Comments to the Author**

Reviewer #1: All comments have been addressed

Reviewer #2: (No Response)

2. Is the manuscript technically sound, and do the data support the conclusions?

Reviewer #1: Yes

Reviewer #2: Partly

3. Has the statistical analysis been performed appropriately and rigorously?

Reviewer #1: Yes

Reviewer #2: N/A

4. Have the authors made all data underlying the findings in their manuscript fully available?

Reviewer #1: Yes

Reviewer #2: No

5. Is the manuscript presented in an intelligible fashion and written in standard English?

Reviewer #1: Yes

Reviewer #2: Yes

Reviewer #1: The authors have addressed all my comments critically and in a very detailed. I suggest the publication of this manuscript

Reviewer #2: I have carefully reviewed the manuscript entitled “Association between sleep duration and chronic lung diseases among Chinese middle-aged and older adults: a cross-sectional study.” Although the study employs a large, nationally representative dataset (CHARLS 2011), I regret to inform you that I do not recommend this manuscript for publication in its current form. My decision is based on several major and minor concerns, as outlined below.

Major Concerns

Study Design and Causal Inference:

The cross-sectional design of the study inherently limits the ability to draw causal inferences between sleep duration and chronic lung diseases (CLDs). While the authors acknowledge this limitation, the discussion does not sufficiently address how this design might influence the interpretation of the observed associations. Future work employing longitudinal data would be more appropriate for establishing causality.

Measurement of Key Variables:

Sleep duration is assessed using self-reported measures that combine nighttime sleep and nap duration. Self-reported data are prone to recall bias and misclassification, potentially impacting the reliability of the exposure measurement. Moreover, the method for categorizing sleep duration (<7, 7–8, and >8 hours) appears somewhat arbitrary and may not capture the nuances of sleep behavior in this population.

Statistical Analysis and Interpretation:

The manuscript uses multiple logistic regression models and a three-piecewise linear regression to explore nonlinearity. However, the rationale for selecting the inflection points at 7 and 8 hours is not convincingly justified by the data or by previous literature. There are also concerns regarding the handling of missing data (using deletion for some variables and dummy coding for others), which might lead to residual confounding. The interpretation of non-significant findings for long sleep duration is ambiguous, and alternative explanations for the observed associations are not fully explored.

Novelty and Contribution to the Field:

While the use of a large dataset is a strength, the overall findings do not present substantial novelty relative to the existing literature on sleep and respiratory health. Several previous studies have reported similar associations or conflicting results. The manuscript would benefit from a more robust discussion that clearly situates its contributions against the current state of research.

Minor Concerns

Clarity and Writing:

The manuscript contains numerous typographical and grammatical errors that detract from the overall clarity. A thorough revision and copy-editing are recommended to enhance readability and ensure the arguments are presented in a clear, concise manner.

Presentation of Data:

Some figures and tables are not well integrated into the narrative. For example, the descriptions of Tables 1–4 and Figure 2 lack sufficient detail in the text. Clear labeling and more comprehensive explanations of the results would improve the presentation.

Ethical and Data Considerations:

Although the manuscript states that the study received ethical approval from the relevant committee, there is limited discussion on the limitations inherent in using self-reported diagnostic information for CLDs. Addressing how potential misclassification might affect the study’s conclusions would be beneficial.

Formatting and Reference Consistency:

Several references and in-text citations appear inconsistent. Please ensure that all references are correctly formatted and that the citations support the claims made in the text.

My Conclusion

In summary, while the research question is of potential interest and the dataset is robust, the methodological limitations, questionable variable measurement, and insufficient justification for the statistical methods substantially weaken the manuscript. I recommend rejection in its current form. Should the authors wish to resubmit, significant revisions addressing these concerns—particularly regarding study design, statistical analysis, and clarity—would be necessary.

**Do you want your identity to be public for this peer review?** For information about this choice, including consent withdrawal, please see our Privacy Policy

Reviewer #1: No

Reviewer #2: No

---

## [Decision Letter · Decision Letter 2]

15 Dec 2025

Dear Dr. Xu,

**Please read the reviewer's concerns about your statistical analysis and address them in your revised manuscript, providing a point-by-point response to the comments upon resubmission.**

plosone@plos.org . A letter that responds to each point raised by the academic editor and reviewer(s). You should upload this letter as a separate file labeled 'Response to Reviewers'.A marked-up copy of your manuscript that highlights changes made to the original version. You should upload this as a separate file labeled 'Revised Manuscript with Track Changes'.An unmarked version of your revised paper without tracked changes. You should upload this as a separate file labeled 'Manuscript'.

We look forward to receiving your revised manuscript.

Kind regards,

Sarah Jose, Ph.D.

Staff Editor

PLOS One

**Journal Requirements:**

Reviewers' comments:

Reviewer's Responses to Questions

**Comments to the Author**

Reviewer #1: (No Response)

2. Is the manuscript technically sound, and do the data support the conclusions?

Reviewer #1: No

3. Has the statistical analysis been performed appropriately and rigorously?

Reviewer #1: No

4. Have the authors made all data underlying the findings in their manuscript fully available?

Reviewer #1: Yes

5. Is the manuscript presented in an intelligible fashion and written in standard English?

Reviewer #1: No

Reviewer #1: (No Response)

**Do you want your identity to be public for this peer review?** For information about this choice, including consent withdrawal, please see our Privacy Policy

Reviewer #1: No

---

## [Author Response · Author response to Decision Letter 3]

23 Dec 2025

All questions have been replied to, see the attachment named "Response to Reviewers".

---

## [Decision Letter · Decision Letter 3]

22 Feb 2026

Dear Dr. Xu,

Thank you for submitting your manuscript to PLOS ONE. After careful consideration, we feel that it has merit but does not fully meet PLOS ONE’s publication criteria as it currently stands. Therefore, we invite you to submit a revised version of the manuscript that addresses the points raised during the review process.

We look forward to receiving your revised manuscript.

Kind regards,

Jahida Gulshan

Academic Editor

PLOS One

Journal Requirements:

Additional Editor Comments (if provided): Please check the comments by reviewer 3 and resubmit your paper after addressing them.

Reviewer #3: This manuscript investigates the association between sleep duration and chronic lung diseases (CLDs) using data from the 2011 baseline survey of the China Health and Retirement Longitudinal Study (CHARLS). The study addresses an important public health question using a large, nationally representative dataset. The sample size is substantial (n = 13,759), and the statistical methods are generally appropriate.

However, several methodological, analytical, and reporting issues should be addressed before the manuscript is suitable for publication.

1. Study Design and Causal Language: The cross-sectional nature of the study limits causal inference. While the authors acknowledge this limitation, parts of the discussion imply directional interpretation. Revise the discussion to avoid causal language

2. CHARLS uses a multistage probability sampling design. The manuscript does not clearly state whether: Sampling weights were applied. Clarify whether survey-weighted logistic regression was used.

3. The manuscript adjusts for hypertension, cardiovascular diseases, diabetes, and depression. Some of these variables may lie on the causal pathway between sleep duration and CLDs, leading to possible overadjustment.

4. Replace “risk” with “odds” throughout the manuscript.

5. Clarify the calculation of sleep duration (conversion of nap minutes to hours).

Reviewers' comments:

Reviewer's Responses to Questions

**Comments to the Author**

Reviewer #1: All comments have been addressed

Reviewer #3:

2. Is the manuscript technically sound, and do the data support the conclusions?

Reviewer #1: Yes

Reviewer #3: Yes

3. Has the statistical analysis been performed appropriately and rigorously?

Reviewer #1: Yes

Reviewer #3: Yes

4. Have the authors made all data underlying the findings in their manuscript fully available?

Reviewer #1: Yes

Reviewer #3: Yes

5. Is the manuscript presented in an intelligible fashion and written in standard English?

Reviewer #1: Yes

Reviewer #3: Yes

Reviewer #1: (No Response)

Reviewer #3: This manuscript investigates the association between sleep duration and chronic lung diseases (CLDs) using data from the 2011 baseline survey of the China Health and Retirement Longitudinal Study (CHARLS). The study addresses an important public health question using a large, nationally representative dataset. The sample size is substantial (n = 13,759), and the statistical methods are generally appropriate.

However, several methodological, analytical, and reporting issues should be addressed before the manuscript is suitable for publication.

1. Study Design and Causal Language: The cross-sectional nature of the study limits causal inference. While the authors acknowledge this limitation, parts of the discussion imply directional interpretation. Revise the discussion to avoid causal language

2. CHARLS uses a multistage probability sampling design. The manuscript does not clearly state whether: Sampling weights were applied. Clarify whether survey-weighted logistic regression was used.

3. The manuscript adjusts for hypertension, cardiovascular diseases, diabetes, and depression. Some of these variables may lie on the causal pathway between sleep duration and CLDs, leading to possible overadjustment.

4. Replace “risk” with “odds” throughout the manuscript.

5. Clarify the calculation of sleep duration (conversion of nap minutes to hours).

**Do you want your identity to be public for this peer review?** For information about this choice, including consent withdrawal, please see our Privacy Policy

Reviewer #1: No

Reviewer #3: No

---

## [Author Response · Author response to Decision Letter 4]

3 Mar 2026

The response has been uploaded as an attachment.

---

## [Editor Report · Decision Letter 4]

4 Mar 2026

Association between sleep duration and chronic lung diseases among Chinese middle-aged and older adults: A cross-sectional study

PONE-D-24-40498R4

Dear Dr. %Xu%,

We’re pleased to inform you that your manuscript has been judged scientifically suitable for publication and will be formally accepted for publication once it meets all outstanding technical requirements.

Kind regards,

Jahida Gulshan

Academic Editor

PLOS One
---

## [Editor Report · Acceptance letter]

PONE-D-24-40498R4

PLOS One

Dear Dr. Xu,

I'm pleased to inform you that your manuscript has been deemed suitable for publication in PLOS One. Congratulations! Your manuscript is now being handed over to our production team.

Kind regards,

on behalf of

Dr. Jahida Gulshan

Academic Editor

PLOS One